# Bring Future Vision: Dynamic Computation Allocation Guided by Lightweight Feature Forecaster

Chao Han[1]   Yijuan Liang[1,2]   Zihao Xuan[3]   Daokuan Wu[4]   Wei Zhang[1]   Xiaoyu Shen[1]

## Abstract

The deployment of large language models (LLMs) in real-world applications is increasingly limited by their high inference cost. While recent advances in dynamic token-level computation allocation attempt to improve efficiency by selectively activating model components per token, existing methods rely on greedy routing—a myopic execute-or-skip mechanism that often leads to irreversible information loss and suboptimal token selection. This paper introduces informed routing, a new paradigm that proactively addresses these issues. The key insight is to assess not only a token's immediate importance but also its recoverability, i.e., how well its transformation can be approximated. To this end, we propose the Lightweight Feature Forecaster (LFF), a small predictive module that estimates a unit's output before routing decisions are made. This enables a flexible execute-or-approximate policy that preserves model fidelity while drastically reducing computation. Extensive experiments show that informed routing consistently achieves state-of-the-art performance across static and dynamic pruning approaches. We further present two practical inference pipelines: a pure-PyTorch implementation and a Triton-based custom operator, that translate these gains into real-world speedups, achieving practical acceleration and consistent improvement across various batch sizes. The code is available in https://github.com/EIT-NLP/informed-routing.

## 1. Introduction

The emergence of large language models (LLMs) has catalyzed breakthroughs across diverse industries (Su et al., 2022; OpenAI et al., 2024; Rozière et al., 2024; Cai et al., 2025; Zheng et al., 2025). Scaling laws have established computational requirements as a primary bottleneck in the development and deployment of LLMs (Kaplan et al., 2020; Su et al., 2024). Therefore, reducing this computational overhead has become a key research objective.

Early work primarily focused on **static pruning** methods, which permanently remove a fixed subset of parameters or components from the model (Han et al., 2016; Ma et al., 2023; Xu et al., 2025). While effective for compression, these approaches fail to exploit the varying importance of tokens during inference. More recently, the observation of diverse token criticality has motivated a shift toward **dynamic computation allocation (DCA)** (Raposo et al., 2024; Jiang et al., 2024; Wu et al., 2025; Shin et al., 2025; Bae et al., 2025), where different tokens undergo different amounts of computation. DCA partitions the model into computational units—ranging from coarse-grained layers to finer-grained sub-layer components (e.g., self-attention blocks and feed-foward network blocks within a single layer (Zhao et al., 2025; Nie et al., 2025)), each equipped with a router. These routers, typically small MLPs, are trained post-hoc to decide whether to execute or skip a unit for each token. In practice, important tokens are routed through most of the model's parameters, while less important ones can skip substantial computation. This flexibility mirrors human language processing, where critical words are analyzed in depth while less informative ones receive only shallow processing (Gong et al., 2024; Fan et al., 2025). However, existing DCA methods are constrained by a paradigm we term **greedy routing**. Routers are trained to make a simple, binary choice: fully execute a computational unit or skip it entirely. Performance recovery is then attempted via lightweight fine-tuning (e.g., LoRA (Hu et al., 2022)). [1]. The decision to skip is based on minimizing the *immediate* performance drop, without considering the long-term consequences. This greedy approach

---

[1]Some works have also explored jointly training the router and the recovery module (Jiang et al., 2024), but this often leads to performance degradation (Zhao et al., 2025).

[1]Institute of Digital Twin, Eastern Institute of Technology, Ningbo, China [2]University of Science and Technology of China, Hefei, China [3]The Hong Kong University of Science and Technology, Hong Kong, China [4]The University of Nottingham Ningbo, China. Correspondence to: Xiaoyu Shen <xyshen@eitech.edu.cn>.

*Proceedings of the 43rd International Conference on Machine Learning*, Seoul, South Korea. PMLR 306, 2026. Copyright 2026 by the author(s).

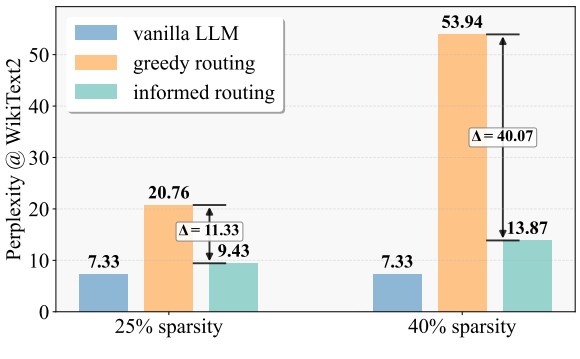

(a) Perplexity After Router Training

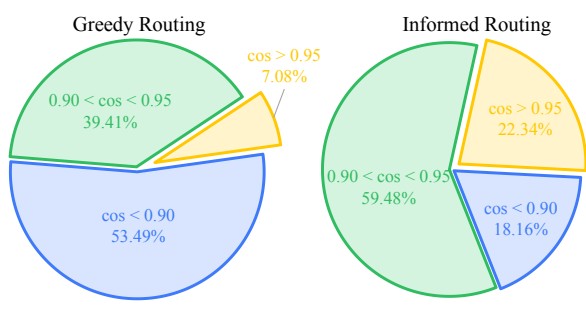

(b) Cosine Similarity Distribution of Features

*Figure 1.* The limitation of greedy routing and the promise of informed routing. Under the same sparity ratio, informed routing (a) reduces the perplexity and (b) increases feature similarity.

suffers from two fundamental flaws:

(1) **The All-or-Nothing Dilemma:** By forcing a rigid execute-or-skip decision, this paradigm offers no middle ground. Skipping a unit causes irreversible information loss, disrupting the model's internal feature distributions and requiring costly fine-tuning to recover performance. As shown in Figure 1a, skipping tokens leads to a significant increase in perplexity, from 7.33 to 20.76 (25% sparsity) and 53.94 (40% sparsity). (2) **Short-Sighted Token Selection:** The router's focus on immediate impact is a poor proxy for true importance. A token that causes a large immediate drop when skipped is not necessarily indispensable; its transformation might be simple and easily recoverable later. Conversely, a token with low immediate impact might be crucial for maintaining subtle, long-range dependencies that are difficult to restore once lost.

To overcome these limitations, we propose a paradigm shift from greedy routing to **informed routing** that replaces the binary execute-or-skip choice with a more nuanced execute-or-approximate decision. The key insight is that rather than discarding information entirely, we can preserve it through approximation. We equip each computational unit with a Lightweight Feature Forecaster (LFF), a small, efficient network trained to mimic the output of its larger counterpart. By this means, tokens no longer face information loss from skipping but instead receive approximate processing. The router's role fundamentally changes from predicting which tokens to drop to identifying which tokens are predictable: tokens with simple, easily-forecasted transformations are routed through the efficient LFF, while those requiring complex processing use the full unit. Our analysis validates this approach. As shown in Figure 1b, LFF increases the proportion of features with high similarity (cosine similarity >0.95) from 7.08% to 22.34%, demonstrating that a substantial fraction of tokens are indeed predictable. We implement this through a simple three-stage pipeline: (1) train LFF to approximate their corresponding units, (2) train

routers to choose between the original unit and its LFF for each token, and (3) perform optional, lightweight LoRA (Hu et al., 2022) fine-tuning to polish the final model. Importantly, we show that we could shrink the original router's intermediate hidden size and reuse the freed capacity to host the LFF, such that *the overall parameter count and computational cost remain identical to standard DCA*, ensuring a fair comparison in both efficiency and resource usage.

Our contributions are as follows: (1) We identify the core limitations of the prevailing greedy routing paradigm in DCA: its rigid all-or-nothing mechanism and its short-sighted reliance on immediate impact as a routing criterion. To address this, we propose informed routing via a Lightweight Feature Forecaster (LFF), replacing skips with efficient approximations guided by token *recoverability*. (2) We conduct extensive evaluations across both static and dynamic pruning settings, demonstrating that informed routing consistently achieves a state-of-the-art performance-efficiency trade-off. Without global fine-tuning, it outperforms greedy routing by over 10%, even exceeding globally fine-tuned greedy routing by 4% with 50% less training cost. When both methods are globally fine-tuned, informed routing delivers up to 20% improvement on individual task, and up to 6% on average across benchmark. We also uncover linguistically grounded insights: routers preferentially approximate attention modules and skip functional tokens (e.g., punctuation, prepositions) while preserving semantic-rich content words. (3) We develop two practical inference pipelines, one with pure-PyTorch and the other with custom Triton kernal, that translate informed routing into tangible wall-clock speedups. The pure-PyTorch implementation achieves practical acceleration for long sequences (e.g., 1.5x speedup at 50% sparsity), while the custom-operator version with Triton overcomes batch and kernel overhead limitations, delivering consistent speedups across sequence lengths and batch sizes (e.g., up to 1.62× at 50% sparsity, batch size 8).

## 2. Related Works

**Static Pruning** Static pruning reduces computation by permanently removing tokens or model components. *Token Pruning* identifies and removes tokens deemed redundant, allowing them to bypass subsequent transformer layers, which are widely used in Vision-Language Models(VLMs) (e.g., SpecVLM (Ji et al., 2025), VisionDrop (Xu et al., 2025)), often relying on attention scores or auxiliary importance estimators. *Parameter pruning* removes fixed structures such as neurons, layers, or blocks, producing smaller models (e.g., SliceGPT (Ashkboos et al., 2024), Shortened-LLaMA (Kim et al., 2024), ShortGPT (Men et al., 2024), LLM-Streamline (Chen et al., 2025a)). Although efficient, such pruning applies uniformly to all tokens, regardless of their importance.

**Dynamic Computation Allocation** Dynamic computation allocation (DCA) leverages the observation that token representations evolve at different rates. Routers, which are often initialized as lightweight classifiers, dynamically assign tokens to execution paths, enabling token-wise computation. Prior work has explored DCA at various levels of sparsity and granularity: fixed per-layer budgets (MoD (Raposo et al., 2024)), global adaptive allocation (D-LLM (Jiang et al., 2024)), and submodule-level routing within layers (SkipGPT (Zhao et al., 2025)). Despite these improvements, all rely on greedy routing, making binary execute-or-skip decisions that risk information loss and short-sighted token selection.

**Error Compensation** Prior works have explored error compensation to mitigate compression-induced accuracy drops. For example, RECAP (Lee et al., 2025) transfers the statistics of pruned channels to adjacent weights. PRUNE&COMP (Chen et al., 2025b) rescales remaining weights offline to compensate for the magnitude gap after layer removal. Olica (He & Lin, 2025) introduces a linear mapping for low-rank compensation in FFNs. ROE (Wu et al., 2025) also identifies the feature gap issue caused by router skipping in multimodal QA tasks, and implements lightweight network adaptation (termed as adapter) at the instance level. While these methods perform *weight-wise*, *channel-wise* or *instance-wise* compensation, our approach operates in a *token-wise* manner. We dynamically route tokens to either the original model or a LFF, enabling finer-grained and adaptive error recovery.

## 3. Methodology

In this section, we introduce **informed routing**, a framework that replaces rigid skip decisions with efficient approximations. Figure 2 provides an overview of the framework.

### 3.1. Preliminaries and Notation

Consider a transformer-based LLM with $L$ layers. For layer $\ell \in \{1, \ldots, L\}$, let $\mathbf{X}^\ell \in \mathbb{R}^{N \times d}$ denote the input token embeddings, where $N$ is sequence length and $d$ is the hidden dimension. Each transformer layer $\ell$ processes the input through two components, i.e. self-attention and feed-forward network(FFN), with pre-normalization and residual connections:

$$\mathbf{X}_{\text{att}}^\ell = \mathcal{A}^\ell\left(\text{Norm}(\mathbf{X}^\ell)\right) + \mathbf{X}^\ell \quad \text{(Self-attention module)}$$
$$\mathbf{X}^{\ell+1} = \mathcal{F}^\ell\left(\text{Norm}(\mathbf{X}_{\text{att}}^\ell)\right) + \mathbf{X}_{\text{att}}^\ell \quad \text{(FFN module)}$$

where $\mathcal{A}^\ell$ is the Multi-head self-attention at layer $\ell$, $\mathcal{F}^\ell$ is the Feed-forward network at layer $\ell$, $\mathbf{X}_{\text{att}}^\ell$ is the Intermediate representation after attention, and $\mathbf{X}^{\ell+1}$ is the Output embeddings serving as input to layer $\ell + 1$.

Following SkipGPT's granularity, we decompose each transformer layer $\ell$ into two *computational units*: $\mathcal{U}^{\ell,\text{SA}}$ (self-attention) and $\mathcal{U}^{\ell,\text{FFN}}$ (feed-forward network). For each unit, a lightweight *router* $\mathcal{R}^{\ell,k} : \mathbb{R}^d \to \mathbb{R}^2$ (where $k \in \{\text{SA}, \text{FFN}\}$) makes token-wise pruning decisions. The router is implemented as a two-layer MLP with bottleneck dimension, i.e $\mathbb{R}^d \to \mathbb{R}^{\lfloor d/4 \rfloor} \to \mathbb{R}^2$.

The router outputs decision logits for each token $\mathbf{x}_i^{\ell,k}$:

$$\mathbf{r}_i^{\ell,k} = \mathcal{R}^{\ell,k}(\mathbf{x}_i^{\ell,k}) \in \mathbb{R}^2 \tag{1}$$

with routing probabilities obtained via softmax:

$$p_i^{\ell,k} = \sigma(\mathbf{r}_i^{\ell,k})_1 = \frac{\exp(\mathbf{r}_i^{\ell,k}[1])}{\sum_{c=0}^{1} \exp(\mathbf{r}_i^{\ell,k}[c])} \tag{2}$$

where class $c = 1$ indicates *preserving precision through original LLM compute unit* and $c = 0$ indicates *lightweight fitting via LFF branch*. Figure 2(c) illustrates this computational flow.

During the forward pass, a hard binary mask $\mathbf{p}^{\ell,k}$ is sampled by applying the $\arg\max$ operation to Gumbel-Softmax logits, producing discrete $0, 1$ values (Jang et al., 2017). In the backward pass, the gradient is estimated using a continuous softmax approximation with temperature $\tau$, enabling differentiable training. The temperature is annealed linearly from $\tau = 5.0$ to $\tau = 1.0$ to sharpen the distribution over time. Modern frameworks such as PyTorch provide built-in functions like $F.gumbel\_softmax$, which facilitates end-to-end training of discrete latent variable models.

The training objective minimizes computation while preserving performance by enforcing a target relative sparsity $S_{\text{target}}$ (e.g., 50%). The global computation fraction is:

$$\rho = \frac{1}{2LN} \sum_{\ell=1}^{L} \sum_{k \in \{\text{SA,FFN}\}} \|\mathbf{p}^{\ell,k}\|_0 \tag{3}$$

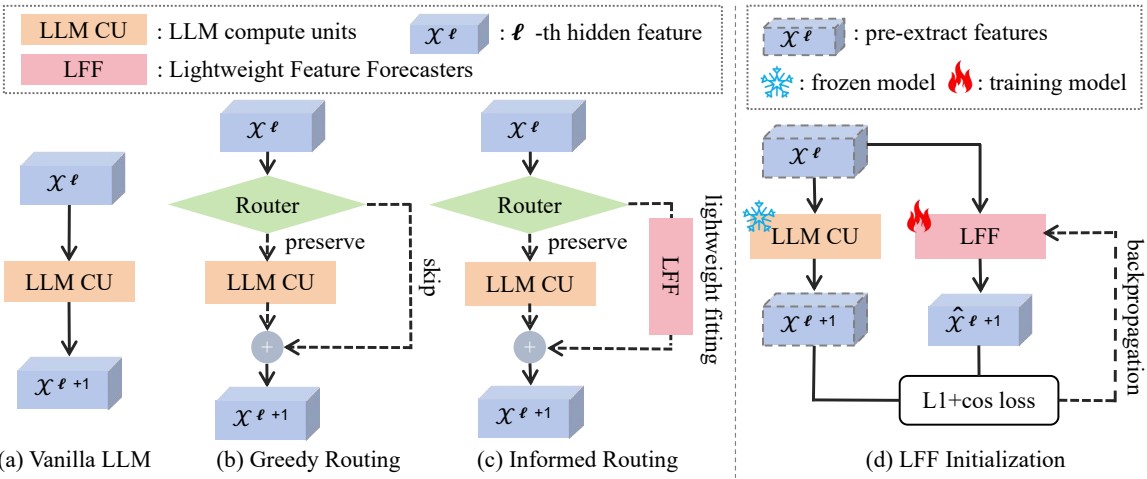

*Figure 2.* (a), (b), and (c) present the architectural comparison diagrams of the vanilla LLM, greedy routing, and our proposed informed routing paradigm. (d) illustrates how the LFF initialization is performed.

where $\|\mathbf{p}^{\ell,k}\|_0 = \sum_i p_i^{\ell,k}$, and we regulate $\rho$ toward $S_{\text{target}}$ during training (Section 3.3).

### 3.2. Lightweight Feature Forecaster

The core innovation of our paper is the **lightweight feature forecaster** $\mathcal{F}^{\ell,k} : \mathbb{R}^d \to \mathbb{R}^d$ that approximates the input-output mapping of computational unit $\mathcal{U}^{\ell,k}$ *before* routing decisions. This architectural shift transitions the routing paradigm from reactive recovery to proactive preservation. For efficiency, $\mathcal{F}^{\ell,k}$ uses a bottleneck architecture:

$$\mathcal{F}^{\ell,k}(\mathbf{x}) = \mathbf{W}_2^{\ell,k} \cdot \left( \mathbf{W}_1^{\ell,k}\mathbf{x} + \mathbf{b}_1^{\ell,k} \right) + \mathbf{b}_2^{\ell,k} \quad (4)$$

where $\mathbf{W}_1^{\ell,k} \in \mathbb{R}^{r \times d}$, $\mathbf{W}_2^{\ell,k} \in \mathbb{R}^{d \times r}$ with $r \ll d$ (e.g., $r = 100$ with $d = 4096$ for Llama3.1-8B (Grattafiori et al., 2024)). This yields minimal parameters: $4096 \times 100 + 100 \times 4096 \approx 0.82$M ($0.02\%$ of $\mathcal{U}^{\text{FFN}}$).

$\mathcal{F}^{\ell,k}$ predicts $\mathcal{U}^{\ell,k}$'s normalized output $\mathbf{z}_i^{\ell,k} \triangleq$ Norm $\left( \mathcal{U}^{\ell,k}(\mathbf{x}_i^{\ell,k}) \right)$ using *cosine similarity loss* and $L_1$ *loss*. Obviously, when the LFF outputs all zeros (i.e., when all its weights are zero), our method degenerates to greedy routing.

### 3.3. Three-Stage Optimization

To effectively implement informed routing, we adopt a three-stage optimization pipeline that separates forecasting, routing, and fine-tuning. By decoupling these objectives, each stage can focus on a specific subtask: feature approximation, routing, or model refinement, allowing for stable and efficient training.

**Stage 1: LFF Initialization.** We commence by training the feature forecasters $\mathcal{F}^{\ell,k}$ to approximate the functional mapping of each computational unit $\mathcal{U}^{\ell,k}$. During this phase, the base LLM parameters remain *frozen*, preserving the original feature distributions. For each unit $(\ell,k)$, we minimize the forecasting loss $\mathcal{L}_{\text{fit}}^{\ell,k}$ using feature pairs $\{(\mathbf{x}_i^{\ell,k}, \mathbf{z}_i^{\ell,k})\}_{i=1}^N$ extracted from a random-selected subset (2,000 samples) of the training corpus.

This decoupled training paradigm (as shown in Figure 2(d)) admits two significant advantages:

1. *Architectural Independence*: Each $\mathcal{F}^{\ell,k}$ learns a *local approximation* of $\mathcal{U}^{\ell,k}$ without gradient propagation between computational units. This isolation eliminates inter-unit dependencies, enabling:

2. *Massive Parallelization*: Forecasters across all $L$ layers and $k \in \{\text{SA}, \text{FFN}\}$ can be trained concurrently via:

$$\underset{\theta_{\mathcal{F}^{\ell,k}}}{\text{minimize}} \, \mathbb{F}_{(\mathbf{x},\mathbf{z})\sim\mathcal{D}} \left[ \mathcal{L}_{\text{fit}}^{\ell,k} \left( \mathcal{F}^{\ell,k}(\mathbf{x};\theta), \mathbf{z} \right) \right] \quad \forall(\ell,k)$$

where $\theta$ denotes forecaster parameters and $\mathcal{D}$ the feature dataset, ensuring computational efficiency.

Feature tensors $\mathbf{X}^{\ell,k}$ and $\mathbf{Z}^{\ell,k}$ can be precomputed offline, circumventing GPU memory bottlenecks associated with full-model activations. For LLaMA3.1-8B (with 64 LFF), this stage completes in less than 5 minutes on a single NVIDIA RTX 6000 Ada GPU (48GB VRAM).

**Stage 2: Router Training.** Next, routers $\mathcal{R}^{\ell,k}$ are trained with both the LLM and LFFs frozen. Each router is a lightweight two-layer MLP:

$$\mathcal{R}^{\ell,k}(\mathbf{x}) = \text{Linear}_{\lfloor d_1 \rfloor \to 2} \left( \text{ReLU} \left( \text{Linear}_{d \to \lfloor d_1 \rfloor}(\mathbf{x}) \right) \right) \quad (5)$$

Although the LFF is already sufficiently lightweight, to ensure a fair comparison with baseline methods, we compromise on the parameter configuration of the router to match the parameter count of SkipGPT. Specifically, the intermediate dimension ($d_1$) of our router is 200 lower than that of SkipGPT. For instance, in the case of the Llama3.1-8B model, SkipGPT adopts an intermediate dimension of $4096/4 = 1024$, while we use $4096/4 - 200 = 824$. As a result, SkipGPT introduces a total of 268.56M parameters (routers), whereas we introduce 268.54M parameters (routers + LFF).

Routers are trained using a composite loss:

$$\mathcal{L}_{\text{route}} = \mathcal{L}_{\text{LM}} + \lambda_1 \mathcal{L}_{\text{sparse}} = \mathcal{L}_{\text{LM}} + ||\rho - S_{\text{target}}||_1 \quad (6)$$

where $\mathcal{L}_{\text{LM}}$ is the language modeling loss, $\rho$ is global compute fraction (Eq. 3) and $\lambda_1 = 8.0$ balances two objective.

**Stage 3: Parameter-Efficient Fine-tuning.** Finally, we optionally inject LoRA adapters into attention projections ($\mathbf{W}_Q, \mathbf{W}_K, \mathbf{W}_V$) and FFN gates:

$$\mathbf{W} \leftarrow \mathbf{W} + \mathbf{AB}, \quad \mathbf{A} \in \mathbb{R}^{d \times r, \mathbf{B} \in \mathbb{R}^{r \times d}}, \ r = 16$$

where $r$ is the rank in lora optimization. Minimizing $\mathcal{L}_{\text{LM}}$ with routers/LFF frozen can further recover performance.

# 4. Experiments

## 4.1. Experimental Setup

Our experimental setup largely follows the configuration used in SkipGPT (Zhao et al., 2025). We evaluate the proposed method on open-source Llama (Grattafiori et al., 2024) models of different scales, specifically 3B (results reported in A.3) and 8B parameters. The RedPajama-Data-1T-Sample (Weber et al., 2024) corpus is employed for both calibration and training. Performance is assessed across two types of tasks: (1) *Reasoning Tasks*, including BoolQ (Clark et al., 2019), PIQA (Bisk et al., 2020), HellaSwag (Zellers et al., 2019), Winogrande (Sakaguchi et al., 2021), ARC-E/C (Clark et al., 2018), and OBQA (Mihaylov et al., 2018), evaluated using the lm-evaluation-harness (Gao et al., 2024); and (2) *Language Modeling Tasks*, measured by perplexity on WikiText-2 (Merity et al., 2017). For comparison, we include state-of-the-art static pruning and dynamic computation allocation methods, with detailed descriptions provided in Appendix A.2.

## 4.2. Results

We conduct extensive experiments to evaluate the proposed informed routing paradigm with LFF. Our results demonstrate its advantages in training stability, efficiency, and performance preservation compared to the traditional greedy routing approaches and static compressing methods. Notably, except for SkipGPT and LFF (ours), all methods

report results from LoRA finetuned models. To further demonstrate the effectiveness of our method, we report two-phase results (router training + LoRA finetune) for SkipGPT and LFF, since SkipGPT essentially serves as an ablation experiment for the informed routing component in our method.

**Inconsistency Between Language Modeling and Reasoning** Experimental results reveal an inconsistency between compressed models' language modeling capability and their reasoning performance. As shown in Table 1, while perplexity (PPL) trends generally align with reasoning accuracy, certain methods deviate. For instance, SliceGPT at 25% sparsity ranks second in PPL but drops to sixth in average reasoning accuracy. Similarly, at 40% sparsity, LFF-Router achieves better PPL than SkipGPT-LoRA yet shows a 3% drop in reasoning. These indicate that LM loss alone may not fully reflect a compressed model's reasoning ability, underscoring the need for evaluation across diverse task-specific benchmarks.

**Training Stability and Efficiency Gains of LFF Initialization** The proposed informed routing approach significantly improves training stability and efficiency by initializing the router with a pre-fit LFF. This leads to faster and smoother router convergence. Intuitively, after router training, at 25% sparsity, LFF-Router reduces PPL by 11 points compared to SkipGPT-Router; at 40% sparsity, the reduction reaches 40 points. Notably, LFF-Router at 25% sparsity outperforms fully fine-tuned SkipGPT-LoRA while saving over 50% training time (details can be found in Appendix A.1).

**Superiority After Fine-tuning and Underlying Mechanisms** After LoRA fine-tuning, our method outperforms SkipGPT in 15 out of 16 tasks. We attribute this to two factors: **(1)** The LFF better preserves the original feature distribution by approximating the layer transformation instead of discarding tokens. Features processed by LFF show higher cosine similarity and lower L1 loss (0.16 vs. 0.52), providing a warmer start for fine-tuning. **(2)** Pre-fitting the LFF enables the router to prioritize tokens with high recoverability—those predictable by a simple network—leading to a healthier model structure and better parameter recovery during LoRA fine-tuning.

## 4.3. Further Analysis

**Which Module to Route, Attention or FFN?** As shown in Table 2, our method consistently select more tokens from self-attention modules than the SkipGPT baseline, across all sparsity levels. This supports prior findings (He et al., 2024) that self-attention is more redundant than FFN blocks. The success of our lightweight, linear LFF in predicting attention outputs suggests that many token transformations in self-attention are approximable by simple linear opera-

*Table 1.* Performance comparison of different pruning methods on reasoning and language modeling tasks at sparsity levels of 25% and 40%. For reasoning tasks, we report accuracy (%); higher is better. The average (AVG) accuracy across all reasoning tasks is included. For Wikitext-2 (WT2), we report perplexity (PPL); lower is better. The best results under each sparsity level are highlighted in **bold** and the second best are underlined.

*(a) Sparsity = 25%*

| Method | Reasoning (Acc. ↑) | | | | | | | | WT2 (PPL ↓) |
|---|---|---|---|---|---|---|---|---|---|
| | BoolQ | OBQA | PIQA | WinoG. | Hella. | ARC-C | ARC-E | AVG | |
| Dense | 82.14 | 44.60 | 81.07 | 77.43 | 81.89 | 57.68 | 84.81 | 72.80 | 7.33 |
| *Static* | | | | | | | | | |
| SliceGPT | **72.39** | 34.40 | 66.70 | 61.56 | 56.96 | 31.48 | 50.08 | 53.37 | 9.22 |
| Shortened-llama | 71.19 | 37.40 | 73.72 | **71.82** | 69.56 | 44.45 | 66.88 | 62.15 | 10.32 |
| ShortGPT | 72.05 | 38.40 | 73.94 | 70.96 | 69.23 | 43.86 | 68.01 | 62.35 | 11.13 |
| *Dynamic* | | | | | | | | | |
| MoD | 50.28 | 31.60 | 64.25 | 52.41 | 50.44 | 28.24 | 37.67 | 44.98 | 34.21 |
| D-LLM | 50.36 | 30.20 | 57.4 | 52.49 | 37.64 | 28.16 | 37.12 | 41.91 | 40.12 |
| SkipGPT-Router | 54.13 | 27.60 | 53.92 | 54.46 | 60.92 | 39.25 | 68.31 | 51.23 | 20.76 |
| LFF-Router (ours) | 71.19 | 40.80 | 74.97 | 63.69 | 73.35 | 49.23 | 79.08 | 64.62 | 9.43 |
| SkipGPT-Lora | 70.67 | 29.60 | 56.96 | 62.83 | 74.22 | 49.91 | 78.79 | 60.43 | 10.53 |
| LFF-Lora (ours) | 71.93 | **41.80** | **76.82** | 65.19 | **76.54** | **51.45** | 79.38 | **66.16** | **8.91** |

*(b) Sparsity = 40%*

| Method | Reasoning (Acc. ↑) | | | | | | | | WT2 (PPL ↓) |
|---|---|---|---|---|---|---|---|---|---|
| | BoolQ | OBQA | PIQA | WinoG. | Hella. | ARC-C | ARC-E | AVG | |
| Dense | 82.14 | 44.60 | 81.07 | 77.43 | 81.89 | 57.68 | 84.81 | 72.80 | 7.33 |
| *Static* | | | | | | | | | |
| SliceGPT | **67.52** | 28.20 | 60.61 | 55.41 | 44.15 | 25.34 | 40.70 | 45.99 | 14.87 |
| Shortened-llama | 65.02 | 32.40 | 68.01 | 64.64 | 57.55 | 33.02 | 53.11 | 53.39 | 17.22 |
| ShortGPT | 65.38 | 32.00 | 68.61 | **67.32** | 58.43 | 35.32 | 53.37 | 54.35 | 18.35 |
| *Dynamic* | | | | | | | | | |
| MoD | 50.28 | 33.00 | 65.56 | 51.38 | 54.01 | 30.20 | 38.09 | 46.07 | 40.42 |
| D-LLM | 50.00 | 31.80 | 58.54 | 51.78 | 48.30 | 26.88 | 44.82 | 44.59 | 52.78 |
| SkipGPT-Router | 53.82 | 31.80 | 60.23 | 54.22 | 46.02 | 28.75 | 52.44 | 46.75 | 53.94 |
| LFF-Router (ours) | 64.43 | 36.00 | 71.87 | 52.17 | 59.95 | 37.71 | 69.99 | 56.02 | 13.87 |
| SkipGPT-LoRA | 66.57 | 37.60 | 70.78 | 56.75 | 65.17 | 42.66 | 72.39 | 58.85 | 14.35 |
| LFF-LoRA (ours) | 65.99 | **38.00** | **73.39** | 58.8 | **69.45** | **43.6** | 72.43 | **60.24** | **11.11** |

*Table 2.* Reduction ratio between Attention and MLP modules at different global sparsity levels.

| Method | 25% Sparsity | | 40% Sparsity | | 70% Sparsity | |
|---|---|---|---|---|---|---|
| | Attention | FFN | Attention | FFN | Attention | FFN |
| SkipGPT | 58.0% | 42.0% | 57.8% | 42.2% | 56.4% | 43.6% |
| LFF | 71.4% | 28.6% | 67.2% | 32.8% | 66.2% | 33.8% |

tions. We term this property **linear simplicity**. Our router, preconditioned by the LFF, learns to identify such tokens, leading to a more explainable sparsity profile.

*Table 3.* Performance comparison between balanced models.

| Method | Reason (Avg. Acc.) ↑ | Mode(PPL) ↓ |
|---|---|---|
| SkipGPT-balance | 58.42 | 11.40 |
| LFF-balance | **63.49** | **9.49** |

**Balanced Computation between Attention and FFN** A potential point of discussion is our treatment of self-attention and FFN modules as equally valid candidates for computation reduction, a design choice inherited from SkipGPT. While self-attention contains fewer parameters, its computational complexity scales quadratically with sequence length, often making it the dominant computational bottleneck in modern long-context large language models. To preemptively address any concern that a direct comparison might be unfair, we conducted a rigorous ablation study. In this experiment, we independently controlled and enforced identical sparsity levels for each module type, i.e., self-attention and FFN, across all layers. This ensures a perfectly equitable comparison of the routing strategies' efficiency on a per-module-class basis. The results, showed in Table 3, demonstrate that our informed routing paradigm consistently achieves superior performance compared to the

greedy routing baseline under these controlled sparsity conditions. This finding robustly confirms that the performance gain of our method is not an artifact of an imbalanced reduction strategy but is intrinsically linked to its preservation of features distributions, validating our core hypothesis.

*Table 4.* Top-5 Highly Routed and Rarely Routed Tokens with Corresponding Skip Counts

| Highly Routed | , : 870 | the : 663 | [space] : 588 | in : 468 | and : 397 |
|---|---|---|---|---|---|
| Rarely Routed | more : 4 | ich : 6 | Destiny : 7 | named : 7 | Eastern : 7 |

**Which Token to Route? Intuitive Visualization of Token Selection for the Lightweight Branch** To analyze which tokens are selected, we sampled a subset of 100k tokens from WikiText-2 and aggregated the total routed (which means approximated by LFF for our method) counts across all layers of the model. Table 4 shows representative examples of most/least frequently skipped tokens and their counts (we omit zero-skipped tokens as they are overwhelming in number). Intuitively, the model tends to skip function words (e.g., conjunctions like "and", prepositions like "in"), punctuation marks (e.g., ",", "."), and other low-information tokens. This aligns with intuition: such tokens contribute minimally to semantic meaning and can be approximated by simple linear projections without significant loss. Our router learns to allocate complex computations to content-rich tokens (e.g., nouns, verbs).

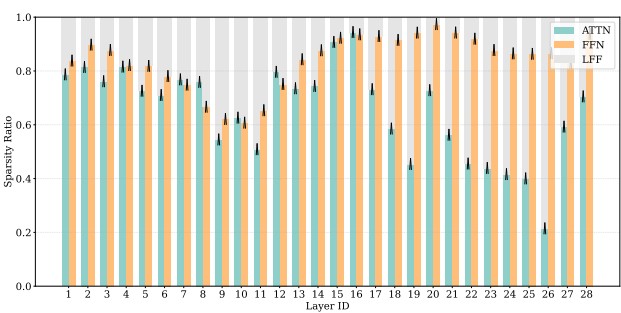

*Figure 3.* Layer-wise Token Allocation. The hatched area represents the proportion of tokens processed by the efficient LFF branch and the colored areas show the tokens retained for full computation in the Attention (green) and FFN (orange) modules.

**Layerwise Allocation Visualization** This section presents an intuitive visualization of token allocation by routers across Attention (ATTN) and Feed-Forward Network (FFN) modules within the Llama3.2-3B model. We track and record the layer-wise allocation information for each batch on the validation set, and then present the average value across all batches. As shown in Figure 3, which details the allocation across 28 layers, the routing mechanism intelligently distributes input tokens, creating a dynamic computational sparsity pattern. A key observation

is that the proportion of tokens directed to the LFF branch (hatched area) varies significantly across layers (also between attention and FFN modules), suggesting that the router adapts its filtering strategy based on the hierarchical processing needs of the network.

**Analysis on Key-Value Cache Reduction** While the primary design of SkipGPT and our method focuses on computational reduction, the memory footprint of the Key-Value (KV) Cache remains a critical bottleneck in autoregressive transformer inference. To directly target KV cache reduction, we adopt an aggressive strategy following (Jiang et al., 2024): an additional masking mechanism is applied to prevent normal tokens from attending to any skipped (or route to LFF branch) tokens in the sequence, to simulate that removing the selected tokens' key-value pairs from the cache.

The performance impact of this operation is non-negligible, as it alters the model's fundamental attention pattern. Our experiments (Table 5) confirm that enforcing this strict KV cache reduction leads to a predictable degradation in model quality. The training convergence loss increases by 0.19, and the perplexity on WikiText2 rises by 2.3 points compared to the standard pruning setup which retains the full cache. However, our LFF still demonstrates performance advantages over the greedy routing baseline.

*Table 5.* Performance of models with/without KV reduction at 25% sparsity

| Method | Validation Loss | FT PPL ↓ |
|---|---|---|
| SkipGPT | 2.42 / 2.55 | 10.53 / 11.68 |
| LFF | 2.30 / 2.49 | 8.93 / 10.21 |

*Table 6.* Performance of models with different FFN - LFFs.

| Version | Avg. Cosine Sim. | PPL(No FT) |
|---|---|---|
| **None** | 0.86 | 36.50 |
| **Linear** | 0.89 | 12.19 |
| **Non-Linear** | 0.89 | 11.99 |
| **SlimmedMLP** | 0.91 | 10.27 |

**Enhanced Version of LFF** As demonstrated in Table 2, the linear LFF exhibits superior fitting capability for attention modules compared to FFN. Naturally, a critical questions arise: *How to design the LFF module appropriately?* To address it, we design two more variants of LFF: 1. **Non-linear**: Incorporating a ReLU activation function in the middle layer to introduce non-linear expressive capacity; 2. **SlimmedMLP**: Adapted from the MLP architecture of LLaMA, with the middle-layer dimension reduced from 12288 to 100 while retaining non-linearity.

Results are listed in Table 6. Intuitively, simply adding a non-linear activation yields nearly identical performance to the fully linear architecture. In contrast, SlimmedMLP yields a notable improvement in the average similarity of feature fitting (from 0.89 to 0.91), with a significant reduction in perplexity (from 12.19 to 10.27). These results suggest that designing LFF by mimicking the structural design of the original large model backbone is a viable strategy. However, this inherently involves a clear trade-off: more parameters and more complex architectures can boost model performance, yet they compromise the acceleration.

**Wall-clock Speed**   Dynamic computation allocation for LLMs is an emerging field without universal acceleration schemes. We thus implement two inference pipelines: (1) Pure-PyTorch: leveraging the router's decisions, it split the token sequence into two groups via `torch.gather`, process them with the full LLM and lightweight branch respectively, then merge them back in original order using `torch.gather` again. (2) Custom-Operator (with triton (Tillet et al., 2019)): It introduces a token-orient block-sparse operator for both attention and feed forward sublayer. Specifically, the operator first sorts the routing mask produced by the router to obtain consecutive indices of the active tokens. Within each thread block, it loads the sorted routing mask and token indices in blocks (e.g., 16, 32, 64, or 128). An online thread block reduction is then performed to check whether the current block contains any tokens that require computation. If so, a vanilla matrix multiplication or flash attention pipeline will be applied sequentially (Dao et al., 2022). Since it only introduces an early-stop mechanism at the block granularity, it remains fully compatible with the original dense kernel, which can deliver actual speedup in an almost free-lunch manner.

*Table 7.* Inference Speed (tokens/s) under Different Sparsity and Sequence Lengths by Pure-Pytorch implementation.

| Sparsity | Seq Len | Prefill | | Decode | |
|---|---|---|---|---|---|
| | | tokens/s | (%) | tokens/s | (%) |
| 0% | 1024 | 1516.36 | – | 1.10 | – |
| 25% | 1024 | 1704.04 | +12.38 | 1.28 | +16.36 |
| 50% | 1024 | 2126.72 | +40.25 | 1.52 | +38.18 |
| 0% | 2048 | 1587.74 | – | 0.62 | – |
| 25% | 2048 | 1742.85 | +9.77 | 0.67 | +8.07 |
| 50% | 2048 | 2254.24 | +41.98 | 0.84 | +35.48 |
| 0% | 4096 | 1464.77 | – | 0.30 | – |
| 25% | 4096 | 1686.27 | +15.12 | 0.34 | +13.33 |
| 50% | 4096 | 2189.87 | +49.50 | 0.46 | +53.33 |

We recorded the inference speed of the proposed algorithm during the prefill and decode phases under different sparsity levels (25%, 50%, with 0% representing the original dense model) and different sequence lengths. Specifically: In the prefill phase, we report the model's token processing speed

*Table 8.* Inference Speed (tokens/s) under Different Sparsity and Batchsize by Custom-Operator implementation.

| spar./bs. | 1 | 2 | 4 | 8 |
|---|---|---|---|---|
| 0% | 1587.74 | 1632.27 | 1659.37 | 1665.18 |
| 25% | 1905.28 | 2072.98 | 2041.02 | 1981.56 |
| 50% | 2476.87 | 2578.98 | 2688.17 | 2697.59 |

(input token length / time consumed); In the decode phase, we report the model's token generation speed.

Under long sequences and high sparsity (e.g., 4096 sequence length and 50% sparsity), Pure-PyTorch implementation achieves significant acceleration, with 1.5x speedup in both prefill and decode phases. However, repeated `torch.gather` operations for token sequence partitioning/merging introduce non-negligible data transfer overhead, which mitigates the acceleration gain—particularly for short sequences and low sparsity regimes. Besides, when $batchsize > 1$, its acceleration effect will deteriorate due to heterogeneous sparsity across sequences, causing batch fragmentation. The block-sparse custom kernel directly accept original dense inputs and dynamically skip computation for inactive blocks on-the-fly based on routing mask, preserves a plug-and-play nature and consistently provides speedup during batch processing. This design achieves consistent acceleration across different batchsize (Table 8), e.g., 1.23x throughout under 25% sparsity and $batchsize = 4$, 1.62x throughout under 50% sparsity and $batchsize = 8$)

## 5. Conclusion

In this work, we identified fundamental limitations in the established greedy routing paradigm for dynamic computation reduction in large language models: its reactive nature leads to irreversible information loss and its token selection criterion is inherently short-sighted. In response, we proposed a paradigm-shifting alternative, informed routing, which introduces Lightweight Feature Forecasters to fit inter-layer transformations before routing decisions are made. Our approach offers key advantages: First, LFFs approximate skipped tokens, reducing feature shift and improving initial stability with less performance drop. Second, LFF forecasting error gives the router a recoverability-based importance measure, enabling smarter retention of hard-to-predict tokens. Third, our method consistently outperform greedy routing methods on both unbalanced and balanced reduction setting. Finally, we show self-attention's redundancy stems from linearly approximable transformations. Future works could explore more sophisticated yet efficient forecasters or hybrid strategies.

## Impact Statement

This paper presents work whose goal is to advance the field of Machine Learning. There are many potential societal consequences of our work, none which we feel must be specifically highlighted here.

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

# A. Appendix

## A.1. Experiment Details

**Training**  Hyper-parameters differ across stages, all stages adopt the same AdamW optimizer (Loshchilov & Hutter, 2017) ($\beta_1 = 0.9$, $\beta_2 = 0.95$).

- *Forecaster Initialization.* Constant learning rate ($1e^{-3}$), training steps (2000), batchsize (8).

- *Router Tuning:* Constant learning rate ($2 \times 10^{-3}$) , training steps (2000), batchsize (16).

- *LoRA Tuning:* Cosine annealing learning rate ($2 \times 10^{-3}$), training steps (2000) with warmup steps (200), batchsize (16).

It is worth emphasizing that in our experiments, we found that computing the router and LFF after normalization (e.g., RMSNorm in llama) can improve training stability—especially when the LFF involves some non-linear activations (e.g., Swish). Therefore, we strongly recommend computing the router and LFF after normalization, which is exactly the approach we adopted in our experiments (to both SkipGPT and LFF).

All experiments were conducted on a single NVIDIA RTX 6000 GPU with 48 GB VRAM. The time consumption of the three experimental phases is summarized in Table 9.

*Table 9.* Time consumption of different experimental phases.

| Experimental Phase | Time Consumption |
| --- | --- |
| LFF Initialization | 5 minutes |
| Router Training | 3 hours |
| LoRA Finetuning | 4 hours |

**Evaluating**  We use lm-eval (Gao et al., 2024) for all evaluation tasks with version 0.4.9. And followed SkipGPT (Zhao et al., 2025), tasks are evaluated with different few-shot contexts, details are listed in Table 10.

*Table 10.* Configuration of few-shot examples and evaluation metrics for different tasks.

| Task Name | Number of Few-shot Examples | Evaluation Metric |
| --- | --- | --- |
| OpenBookQA | 0 | acc_norm |
| Winogrande | 5 | acc |
| PIQA | 0 | acc |
| HellaSwag | 10 | acc_norm |
| BoolQ | 0 | acc |
| ARC-Easy | 25 | acc_norm |
| ARC-Challenge | 25 | acc_norm |
| WikiText2 | 0 | word_perplexity |

## A.2. Comparison Methods

To provide a comprehensive evaluation, the proposed method is compared with several state-of-the-art approaches in static model compression and dynamic computation allocation.

- **SliceGPT** (Ashkboos et al., 2024): This method applies Principal Component Analysis (PCA) on orthogonally transformed parameters to remove entire rows and columns, achieving static parameter pruning. It results in a uniformly smaller model by permanently removing fixed structural components.

- **Shortened-llama** (Kim et al., 2024): This approach focuses on depth pruning by removing consecutive layers in LLMs to create a smaller model. It demonstrates that reducing model depth can be an efficient strategy for LLM inference.

- **ShortGPT** (Men et al., 2024): Leveraging Block Influence (BI), ShortGPT quantitatively estimates the importance of layers to prune less critical ones. This method is a static layer-pruning technique that aims to reduce model capacity.

- **Mixture-of-Depths (MoD)** (Raposo et al., 2024): MoD is a dynamic computation allocation method that enforces a fixed sparsity ratio per layer block. It employs a greedy routing paradigm where routers decide to execute or skip computational units for tokens.

- **D-LLM** (Jiang et al., 2024): This method introduces global adaptive sparsity, dynamically allocating computation across layers based on input characteristics. It refines dynamic computation by allowing more flexible computation paths across the model.

- **SkipGPT** (Zhao et al., 2025): A prominent dynamic computation allocation baseline, SkipGPT further refines granularity by decoupling attention and MLP operations within each layer. It uses separate routers to independently skip sub-modules, operating under the greedy routing paradigm.

### A.3. Generalization on Llama-3B

*Table 11.* Performance comparison on Llama3.2-3B with 25% sparsity. Accuracies (%) on reasoning tasks; perplexity (PPL) on WikiText-2.

| Method | Reasoning (Acc. ↑) | | | | | | | | WT2 (PPL ↓) |
|---|---|---|---|---|---|---|---|---|---|
| | BoolQ | OBQA | PIQA | WinoG. | Hella. | ARC-C | ARC-E | AVG | |
| Dense | 73.03 | 43.40 | 77.58 | 72.22 | 76.41 | 50.85 | 79.17 | 67.52 | 9.27 |
| SkipGPT-Router | 47.65 | 34.40 | 61.15 | 54.14 | 51.99 | 26.62 | 39.56 | 45.07 | 36.50 |
| LFF-Router (ours) | 61.74 | 34.80 | 68.28 | 58.33 | 62.36 | 40.53 | 71.04 | 56.73 | 12.19 |
| SkipGPT-LoRA | 62.81 | 36.60 | 66.49 | **59.19** | 64.52 | 39.51 | 70.71 | 57.12 | 14.82 |
| LFF-LoRA (ours) | **62.87** | **37.20** | **69.04** | 59.04 | **66.14** | **41.81** | **73.06** | **58.45** | **11.46** |

To validate the generalization across model scales, we evaluate our method on the Llama3.2-3B model. As shown in Table 11, the results align with those from the 8B model, substantiating our approach's efficacy. At 25% sparsity, LFF-Router significantly outperforms SkipGPT-Router, improving the average accuracy on reasoning tasks by 11% and reducing language modeling perplexity by 24. Moreover, LFF-Router matches the performance of SkipGPT-LoRA while saving over 50% in training time. After LoRA fine-tuning, LFF-LoRA achieves superior performance on 8 out of 9 tasks, confirming the advantage of informed routing. An key finding is that the performance degradation is more pronounced on the 3B model, indicating its lower intrinsic redundancy and higher sensitivity to computation reduction.

