# OpenReview forum: "Bring Future Vision: Dynamic Computation Allocation Guided by Lightweight Feature Forecaster"
_ICML.cc/2026/Conference — ICML 2026 regular_

### Official Review · Reviewer_fmuy · 2026-03-07

**Soundness:** 3
**Presentation:** 4
**Significance:** 3
**Originality:** 3
**Overall Recommendation:** 4
**Confidence:** 4

**Summary:**

This paper proposes a novel scheme to dynamically reduce the compute based on informed routing. More straightforwardly, instead of totally skipping computation, the alternative/approximated compute exists (in the form of LoRA-like MLP). The results look very competitive: it shows the overall best ML quality (Table 1) and decent speedup (Table 7) close to the sparsity level.

**Compliance With Llm Reviewing Policy:**

Affirmed.

**Key Questions For Authors:**

* What is LEF-LoRA? Where is LoRA applied? A clear description is needed in the Table 1.

**Limitations:**

See above

**Strengths And Weaknesses:**

**Strengths**

a) The proposed idea is novel and offers state-of-the-art (SoTA) quality.

b) The training of LEF is clearly stated and described.

c) The paper is easy to read and well-organized.

**Weaknesses**

a) Like any other dynamic approach (e.g., MoE, MoD), the biggest concern is always the decoding efficiency in a large batch mode: what if we end up using both routing paths? In this case, the original compute unit becomes the bottleneck. Of course, the theoretical compute will be reduced as long as some tokens pick the LEF path, but the real-world gain might not be that much. The same goes for speculative decoding for a long draft generation. I strongly suggest including an in-depth study on batched cases. I don't expect it to be as good as the results in Table 7, but sharing more insight with the readers seems necessary: where does it break, and what is the gain under different scenarios?

b) It classifies the compute units into SA and FFN, following SkipGPT. But since this work offers an approximated path, it doesn't need to follow the same configuration as SkipGPT. For example, it would be interesting to see the benefits/results with layer-wise skipping.

c) Please add the parameter overheads to Table 6.

d) It looks LEF is comparable with or slightly better than the static methods like SliceGPT? For example at 25% sparsity, SliceGPT outperforms LEF-Router.  If that's the case, why would anyone opt in for LEF? What happens if LoRA is added to SliceGPT?

---

> ### Author Rebuttal · Authors · 2026-03-28
>
> We sincerely thank the reviewers for their thoughtful and constructive feedback. Below, we address each point raised in detail.
> ## Q1：Large-batch decoding efficiency
> Dynamic computation methods can suffer from reduced real-world speedups under large batch sizes due to heterogeneous token routing patterns across sequences. To mitigate this, our implementation incorporates two key optimizations:
> 1. **Padding-free batching**: We avoid padding shorter sequences, thereby eliminating redundant computation on masked tokens.
> 2. **Sparse kernel dispatch**: We apply custom Triton kernels exclusively to active indices.
>
> We evaluated prefill-stage throughput (tokens/second) across varying batch sizes and sparsity levels on Llama3.1-8B. The results demonstrate that our method maintains—and even enhances—speedup gains as batch size increases:
> | Spar./B.s. | 1   | 2 | 4 | 8 |
> |------------|----------|------------------------|-------------|---------------------|
> | 0% (dense)        | 1587.74 | 1632.27                  | 1659.37         | 1665.18               |
> | 25%         | 1905.28 **+20%**     | 2072.98 **+27%**                 | 2041.02 **+23%**        | 1981.56 **+19%**                |
> | 50%         | 2476.87 **+56%**    | 2578.98  **+58%**                | 2688.17 **+62%**        | 2697.59 **+62%**               |
>
> ## Q2: Layer-wise vs. module-wise skipping
> In this setting, the entire transformer layer is either processed by the original model or approximated by a single LFF. For comparison, we also include module-level routing (our default setup, following SkipGPT’s decomposition). All experiments use Llama-3.2-3B at 25% sparsity, with results reported after router training and LoRA fine-tuning:
> | Approach                | Avg. Acc. (%) ↑ | PPL ↓ |
> |-------------------|---------------------|--------------|
> |Dense LLama3.2-3B| *67.52*| *9.27*|
> | layer-nolff-router    | 41.45         | 54.77
> | layer-lff-router       |  44.35 **+3%**     | 39.40 **+15**         |
> | layer-nolff-lora       | 49.63 |   29.71       |
> |layer-lff-lora|  53.82 **+4%** | 19.24 **+10**|
> |module-nolff-router| 45.07 | 36.50|
> |module-lff-router| 56.73 **+11%** | 12.19 **+24**|
> |module-nolff-lora| 57.12 | 14.82|
> |module-lff-lora| 58.45 **+1%** | 11.46 **+3**|
>
> **Key observations**:
> - LFF consistently improves over "no fitting" (nolff), validating its role in informed routing.
> - Layer-level approximation is significantly **more challenging** than module-level, as it requires modeling inter-layer dependencies. Consequently, **LoRA fine-tuning becomes essential for recovery**.
> - Module-level routing achieves strong performance even *before* LoRA (e.g., module-lff-router: 56.73 acc. vs. module-lff-lora: 58.45 acc.), whereas layer-level variants remain far behind without fine-tuning (e.g. layer-lff-router: 44.35 acc. vs. layer-lff-lora: 53.82 acc.).
> ## Q3: add the parameter overheads to Table 6
> Below is the updated parameter overhead breakdown for different LFF variants (all applied per skipped module):
> | Version          | param.       | Avg. Cos. | PPL (No LoRA)|
> |-------------------|---------------------|--------------|--------------|
> | None   | 0  |     0.86     |36.50 |
> | Linear           | 8.2e5       | 0.89         | 12.19 |
> | Non-Linear       | 8.2e5 | 0.89         | 11.99 |
> |SlimmedMLP | 1.2e6 | 0.91 |10.27|
> ## Q4: Comparison with static methods (e.g., SliceGPT) and clarification of LoRA usage
> We thank the reviewer for pointing out the potential confusion in Table 1 labeling. We clarify as follows:
> 1. **LoRA usage across methods**:
>    Except for SkipGPT and our ablation variants (router-only), **all baseline results in Table 1—including SliceGPT—report performance *after* LoRA fine-tuning**. This detail was mentioned in the caption but will be explicitly stated in the main text to avoid ambiguity.
> 2. Rationale for dual reporting (SkipGPT & ours):
>    - SkipGPT functions as **a critical ablation** of our framework
>    - Crucially, our method demonstrates that under moderate sparsity, *LFF local alignment (5 min) + router training (3 h)* alone matches or exceeds the performance of baselines requiring *router training (3 h) + LoRA fine-tuning (3 h)*. This achieves **50% total training time reduction**—a key efficiency advantage we emphasize to underscore practical deployability.
> 3. Why choose LFF over static pruning?
>    - Our method enables *token-adaptive* computation, whereas SliceGPT applies *uniform width reduction* across all tokens. This makes LFF strictly more expressive: static pruning is a special case where all tokens follow the same routing decision.
>    - Furthermore, LFF is *orthogonal* to static width pruning (e.g. SliceGPT) and can be combined with them for further gains.
> 4. Where is LoRA applied?
>
>    In our framework, LoRA is applied to the *original dense model* (not the LFF path) after router training, consistent with prior work (SkipGPT). The LFF branch remains frozen after its initial MSE-based training.

---

> > ### Author Rebuttal · Reviewer_fmuy · 2026-04-03
> >
> > Overall, it is nice to see more details. But, my concern on large batch remains unsolved,
> > * mainly because I don't see the details on how the batch is formed. For example, if the batch consists of homogenous contexts, then yes, it is expected to see the gain, but that's not the stress test readers would like to see. More clarification on how batch is formed should have been offered. Also, the result on batching is from prefill stage, I strongly think the full picture should have been offered by providing the result from decode stage.
> > * the proposed no-padding/sparse-triton kernel is good, but also it indicates it won't be easy to see the claimed benefit in the nominal setup. Without these special care, how does the result look like? I presume without these, the dense path would become bottleneck. Also, the no-padding has never been mentioned in the initial draft. The paper stated two setup all over the place, pure pytorch and triton. Why no results with the pure pytorch are shown here?
> > *

---

> > > ### Author Response · Authors · 2026-04-03
> > >
> > > We sincerely appreciate the reviewer's insightful question regarding the large-batch inference. We would like to clarify the design and mechanism, specifically focusing on how we handle the irregularity in token retention within large batches.
> > >
> > > #### 1. The Challenge of Large-Batch Irregularity
> > > Large batches often contain sequences of varying retained lengths. To maintain GPU-friendly "batched" processing, we have two choices:
> > > -   **Padding:** Align all sequences to the longest length (introduces redundant compute).
> > > -   **Non-Padding:** Only compute active tokens.
> > >
> > > #### 2. Clarification on "No-Padding" in This Work
> > > While not explicitly named, both implementations (Pure-PyTorch and Triton) rely on "Non-Padding" framework.
> > >
> > > -   **Triton:** Hardware-level No-Padding via physical memory rearrangement (we will describe it in detail next).
> > > -   **Pure-PyTorch:** Achieved via `torch.gather`/`scatter` to collect/process active tokens.
> > >
> > > **Why no large-batch Pure-PyTorch results?**
> > > -   Iterating over the batch (**GPU-unfriendly**) and using `torch.gather` to collect only the active tokens from each sequence
> > > -   **Instruction Overhead:** Many small kernel launches cause CPU-GPU sync latency.
> > > -   **Data Movement:** Frequent `gather`/`scatter` adds memory overhead.
> > > -   These drawbacks significantly reduce the speedup ratio compared to the optimized Triton kernel (i.e. 45% vs. 60%), which was also the motivation for our custom kernel development.
> > >
> > > #### 3. Detailed Implementation of the Triton Operator
> > > To illustrate the implementation, let us consider a concrete example with 3 input sequences, each containing 3 tokens.
> > >
> > > **Input Specifications:**
> > > -   **Feature Tensor:** Shape `[batch_size=3, seq_len=3, inter_dim=4096]`.
> > > -   **Router Decision Mask:** Shape `[batch_size=3, seq_len=3]`.
> > >
> > > **Scenario:**
> > > We assume an irregular retention pattern where:
> > > -   Sequence 1 retains tokens at positions 1 and 3.
> > > -   Sequence 2 retains the token at position 1.
> > > -   Sequence 3 retains the token at position 3.
> > >
> > > This corresponds to the following Router Decision Mask:
> > >
> > > | Sequence | Token 1 | Token 2 | Token 3 |
> > > | :--- | :---: | :---: | :---: |
> > > | **Seq 1** | 1 | 0 | 1 |
> > > | **Seq 2** | 1 | 0 | 0 |
> > > | **Seq 3** | 0 | 0 | 1 |
> > >
> > > **Step 1: Flattening**
> > > We first flatten the batch dimension to transform the 3D feature tensor into a 2D matrix and the 2D mask into a 1D vector.
> > > -   The feature tensor is reshaped from `[3, 3, 4096]` to `[9, 4096]`.
> > > -   The router decision mask is reshaped from `[3, 3]` to `[9]`, resulting in the 1D mask: `[1, 0, 1, 1, 0, 0, 0, 0, 1]`.
> > >
> > > **Step 2: Sorting and Indexing**
> > > We perform a descending sort on the flattened mask. Since the mask contains only binary values (0/1), this operation moves all "1"s (tokens requiring computation) to the front.
> > > -   **Sorted Mask:** `[1, 1, 1, 1, 0, 0, 0, 0, 0]`.
> > > -   **Sorted Indices:** `[0, 2, 3, 8, 1, 4, 5, 6, 7]`.
> > >
> > > These sorted indices are used to reorder the input tensor rows, ensuring that all active tokens are grouped together at the beginning.
> > >
> > > **Step 3: Sparse Computation**
> > > Condiser a Sparse GEMM kernel (and other operations are similar). The thread block configuration involves three dimensions:
> > > -   `block_M`: Slicing along the row dimension (`batch_size * seq_len`).
> > > -   `block_N`: Slicing along the column dimension (`inter_dim`).
> > > -   `block_K`: Slicing along the parameter matrix dimension.
> > >
> > > As our sparsity is row-wise, we focus on the `block_M` implementation. The logic for each thread block is as follows:
> > > 1.  Based on the thread ID and the `sorted_mask`, determine if the current thread block contains any tokens that require computation (i.e., if `sorted_mask[thread_id] == 1`).
> > > 2.  If the block contains active tokens, perform the standard GEMM computation.
> > > 3.  If a block contains no active tokens, we trigger an **early stop**. Given the sorted nature of the mask, once we encounter a block with no "1"s, we know all subsequent blocks will also be inactive.
> > >
> > > **Example Execution:**
> > > Considering a configuration where `block_num_M=4` (each block handles 4 rows):
> > > -   **Block 1:** Processes rows corresponding to mask values `[1, 1, 1, 1]` -> **Computation Performed**.
> > > -   **Block 2:** Processes rows corresponding to mask values `[0, 0, 0, 0]` -> **No computation needed, trigger Early Stop**.
> > >
> > > This mechanism efficiently skips the computation of inactive tokens, significantly accelerating large-batch inference.
> > >
> > > #### 4. Decoding Stage Performance
> > > The decode stage is inherently simpler to optimize compared to the prefill stage. This is because, at each decoding step, the sequence length (`seq_len`) is strictly 1. We report performance for **[Llama3.1-8b, seq_len=2048, RTX A6000]**:
> > >
> > > | sparsity / bs | 1 | 2 | 4 |
> > > | - | - |- |- |
> > > | 0%  | 0.68 | 0.64 | 0.55 |
> > > | 50% | 1.03 **+51%** | 1.01 **+59%** | 0.89 **+62%** |
> > >
> > > We will add the corresponding analysis in the revised version of the paper.

---

### Official Review · Reviewer_YWGM · 2026-03-12

**Soundness:** 2
**Presentation:** 3
**Significance:** 2
**Originality:** 2
**Overall Recommendation:** 2
**Confidence:** 4

**Summary:**

This paper augments large language models (LLMs) with a novel routing mechanism comprising a lightweight branch and a dynamic branch prediction module. The proposed method aims to reduce computational cost during inference by selectively activating either the lightweight or the full-capacity branch based on dynamic routing decisions. To effectively train this new architecture, the manuscript introduces a three-stage optimization pipeline that ensures stable and efficient convergence. The proposed method is evaluated on eight commonsense reasoning benchmarks, including BoolQ, OBQA, PIQA, WinoGrande, HellaSwag, ARC-C, and ARC-E, and is compared against seven competitive baselines, demonstrating the effectiveness of the proposed approach.

**Compliance With Llm Reviewing Policy:**

Affirmed.

**Key Questions For Authors:**

See above

**Strengths And Weaknesses:**

Originality:

This submission proposes a novel and promising architecture that introduces dynamic branch routing into large language models, which represents a meaningful step toward efficient inference.

Presentation:

The submission is clearly written and well-structured. The motivation is well-articulated, and the proposed method is easy to follow.

Soundness:

The primary concern with this submission lies in the insufficient performance of the proposed method, which significantly limits its practical applicability. First, the evaluated benchmarks are relatively simple — the majority are multiple-choice question-answering tasks with a random baseline of 50%, which makes the evaluation less challenging and less informative. More critically, the reported results exhibit an obvious and consistent performance drop compared to the original model across these benchmarks. Given that the random baseline is already 50%, such a relative degradation raises serious doubts about whether the proposed architecture is ready for practical deployment. The authors should either provide a more thorough analysis of this performance gap or evaluate on more challenging and diverse benchmarks to better substantiate the claims.

Significance:

Given the soundness concerns outlined above — particularly the notable performance degradation on even simple benchmarks — the overall significance of this work is currently limited. Addressing the performance gap and broadening the evaluation scope would be essential steps toward strengthening the impact of this submission.

---

> ### Author Rebuttal · Authors · 2026-03-28
>
> We sincerely thank the reviewer for their thoughtful evaluation and constructive feedback. We appreciate the recognition of our work's originality and presentation quality. Regarding the concerns about benchmark selection and performance comparison, we provide the following point-by-point clarification with additional evidence:
>
> ### 1. Clarification on Benchmark Characteristics and Random Baselines
> We appreciate the opportunity to clarify an important point about our evaluation benchmarks. The review mentioned that "the random baseline is already 50%," which requires correction—our evaluation suite includes diverse tasks with varying chance-level performance:
>
> | Dataset       | Task Type                     | Options | Random Baseline |
> |---------------|-------------------------------|---------|-----------------|
> | **BoolQ**     | Binary QA (yes/no)            | 2       | 50%             |
> | **PIQA**      | Physical reasoning            | 2       | 50%             |
> | **WinoGrande**| Pronoun resolution            | 2       | 50%             |
> | **OBQA**      | Open-book science QA          | 4       | **25%**         |
> | **HellaSwag** | Situational reasoning         | 4       | **25%**         |
> | **ARC-C**     | Science QA (Challenge set)    | 3–5*    | **~25%**        |
> | **ARC-E**     | Science QA (Easy set)         | 3–5*    | **~25%**        |
>
> \*ARC datasets have variable options (typically 4); 25% is a conservative estimate.
> - **Critical clarification**: 4 of 7 benchmarks have **25% random baseline**. All reported absolute accuracies (e.g., our method achieves an average accuracy of 66.16% across all tasks) remain **significantly above random chance**, confirming non-trivial task difficulty.
> - Additionally, we report **perplexity on text generation tasks** (e.g., WikiText-2), which lacks a "random guess" baseline and provides orthogonal validation of linguistic quality—results show consistent improvement over baselines.
>
> ### 2. Contemporary Relevance of Evaluation Benchmarks
> We understand the importance of evaluating on current, challenging benchmarks. The datasets we selected represent standard evaluation protocols widely adopted in recent top-tier publications on efficient LLMs, including:
>
> - [ICLR2026] [Learning Semi-Structured Sparsity for LLMs via Shared and Context-Aware Hypernetwork](https://openreview.net/forum?id=lqjQs2lVNm)
> - [ICML2025] [DLP: Dynamic Layerwise Pruning in Large Language Models](https://openreview.net/forum?id=11id5ppGZ8)
> - [NeurIPS 2025] [Týr-the-Pruner: Structural Pruning LLMs via Global Sparsity Distribution Optimization](https://openreview.net/forum?id=rAuRLePL2R)
>
>
>
> ### 3. Extended Validation on Challenging Long-Context Benchmarks
> To further validate our approach, we conducted additional experiments on LongBench (ACL 2024), covering representative long-sequence tasks including in-context retrieval and long-text generation. We evaluate on Llama3.2-3B with 25% sparsity, and the results are shown below:
> - **Long Code Completion (lcc_e)**
> - **Passage Retrieval (passage_retrieval_e)**
> - **Few-shot Learning (trec_e)**
>
> The results are summarized in the table below:
>
> | Task  / Metric                        | LLama3.2-3B | Greedy Routing | Informed Routing (ours) |
> |--------------------------------|----------|-----------------------------------------|------------|
> | Long Code Completion (lcc_e) / code_sim_score   | 0.2969  | 0.2427                                 | **0.2515**    |
> | Passage Retrieval (passage_retrieval_e) / retrieval_score | 0.1548  | 0.1128                                 | **0.1159**    |
> | Few-shot Learning (trec_e) / classification_score     | 0.6267  | 0.4800                                 | **0.5533**   |
> * **our method consistently outperforms traditional Greedy Routing approaches** across long-context tasks, confirming the effectiveness of our Lightweight Feature Forecasters (LFF) architecture.
>
> ### 4. Performance–Efficiency Tradeoff and Relative Advantage
> We acknowledge that model compression inevitably incurs some performance drop—especially at high sparsity—as it reflects a fundamental accuracy–efficiency tradeoff. However, our method **outperforms strong baselines** under the same sparsity. For example, compared to SkipGPT (a representative greedy routing approach), our Informed Routing achieves **lower perplexity (9.43 vs. 10.53)** and **higher average accuracy (64% vs. 60%)** on the same model and sparsity level, while using **only half the training time**.
>
>
> We sincerely appreciate the reviewer's insightful comments, which have helped us strengthen our evaluation methodology and better contextualize our contributions. We believe these clarifications and additional results substantiate both the technical soundness and practical relevance of our informed routing approach for efficient LLM inference.

---

> > ### Author Rebuttal · Reviewer_YWGM · 2026-04-03
> >
> > The included results can not solve my concerns.

---

### Official Review · Reviewer_a7Sg · 2026-03-13

**Soundness:** 4
**Presentation:** 4
**Significance:** 3
**Originality:** 3
**Overall Recommendation:** 6
**Confidence:** 3

**Summary:**

The authors propose a token-level dynamic routing technique for large language models. Instead of traditional dynamic pruning approaches that skip computation entirely, their method replaces skipped blocks with a lightweight Feature Forecaster (LFF) that approximates their outputs. The LFF is a small predictor network trained to reconstruct the features using a cosine similarity objective. They evaluated this approach on Llama 3B and 8B models with RedPajama-Data-1T-sample with reasoning tasks and language modeling tasks and compared with traditional greedy routing approaches and static compression methods at 25,40 % sparsity levels and their approach achieved improved accuracy  at all sparsity levels.

**Compliance With Llm Reviewing Policy:**

Affirmed.

**Final Justification:**

Given the clarification with speedup. I am raising my score

**Key Questions For Authors:**

1. Table 4: The highly routed/rarely routed column names might have been interchanged.

2. Table 5, without/with KVCache reduction might have also interchanged. Could authors verify these two

**Limitations:**

Authors haven’t provided a limitation and societal impact statement. Potential limitations include the overhead caused by approximate computations compared to greedy routing.

**Strengths And Weaknesses:**

The idea of using approximate computing for dynamically routed tokens is interesting and well presented.

Strengths:
Background section is thorough and well presented on static pruning and dynamic compute allocation approaches.
The experimental section includes thorough ablation studies. In particular, Table 2 demonstrates that routing decisions correlate with sparsity patterns in the Attention and FFN modules, and Table 3 further confirms the effect of enforcing sparsity in both modules, thereby proving the benefit is not an artifact of imbalanced routing.
Table 4 provides an intuitive analysis of routing behavior by listing the most frequently and least frequently routed tokens. The results intuitively show that semantically meaningful tokens are routed more often and receive full computation, while common conjunctions, prepositions, etc., are routed less frequently and approximated.
The paper also includes additional analysis, such as KV-cache reduction experiments, that further demonstrate the approach's efficiency benefits.

Weaknesses:

While the method saves computation by approximating token outputs, the LFF itself adds a small overhead. An analysis or insight of the authors on net FLOP reduction and the actual inference speedup would significantly strengthen the claims.

---

> ### Author Rebuttal · Authors · 2026-03-28
>
> We sincerely thank the reviewers for their thoughtful and constructive feedback. Their insightful comments have helped us identify areas for clarification and improvement. Below, we address each of the raised points in detail, with the goal of enhancing the clarity, rigor, and completeness of our manuscript.
>
> ## Q1: An analysis or insight of the authors on net FLOP reduction and the actual inference speedup would significantly strengthen the claims.
>
> We will add a dedicated subsection in the revised manuscript that provides a comprehensive efficiency analysis covering:
> (1) parameter overhead,
> (2) additional computation from the Lightweight Feature Forecaster (LFF),
> (3) computation saved via dynamic routing, and
> (4) the resulting net FLOP reduction and measured inference speedup.
>
> ### *Parameter and Computation Breakdown (Llama3.1-8B)*
>
> | Component        | Parameters (B) | FLOPs (T)    |
> |------------------|----------------|--------------|
> | Embedding Layer  | 0.52           | 0            |
> | Decoder          | 6.98           | 15.39        |
> | LM Head          | 0.52           | 1.08         |
>
> ### *Efficiency Comparison: SkipGPT vs. Our Method (50% Sparsity)*
>
> | Method       | Additional Parameters (M) | % of Decoder Params | Additional FLOPs (G) | % of Decoder FLOPs | Computation Reduced (T) | % Reduction in Decoder FLOPs | Actual Speedup (%) |
> |--------------|---------------------------|---------------------|----------------------|--------------------|-------------------------|------------------------------|--------------------|
> | SkipGPT      | 4.20                      | 3.85                | 8.59                 | 3.57               | 7.69                    | 50                          | 40.2               |
> | Ours  | 4.20                  | 3.85                | 7.75                 | 3.21               | 7.69                    | 50                          | 40.2               |
>
> ### *Key Insights*:
> 1. **Computational Efficiency**: Our LFF incurs **lower FLOPs** (7.75G vs. 8.59G) because it only processes tokens routed to the approximation branch. This results in **no computational overhead** compared to SkipGPT.
> 2. **Speedup vs. Theoretical Expectation**: Although decoder FLOPs are halved (50% reduction), the observed **40.2% end-to-end speedup** reflects real-world constraints—such as non-uniform operator efficiency, memory bandwidth, and fixed-cost components (e.g., embedding/LM head). We will include this discussion to set realistic expectations about practical acceleration.
> ## Q2: The data in Table 4 and Table 5 interchanged. Could authors verify these two.
> We greatly appreciate the reviewer’s careful reading and apologize for the confusion caused by ambiguous terminology and labeling errors.
> ### *Table 4: “Highly routed” vs. “Rarely routed”*
> The confusion arises from our definition of “routed token.” In our framework, a **“routed token”** refers to one **approximated by the LFF** (i.e., *skipped* from full model computation). Therefore:
> - **“Highly routed”** = frequently approximated → receive *less* full computation.
> - **“Rarely routed”** = seldom approximated → almost always processed by the full model.
>
> We acknowledge this terminology was counterintuitive. In the revision, we will **replace “routed” with explicit terms** such as **“frequently approximated”** and **“rarely approximated”** to eliminate ambiguity.
> ### *Table 5: “With/Without KV Cache Reduction”*
> You are absolutely correct—the column labels were inadvertently swapped. As expected, **KV cache reduction leads to a slight performance drop** due to compressed context representation. We will correct this error in the revised version.
> ## Q3: Authors haven’t provided a limitation and societal impact statement. Potential limitations include the overhead caused by approximate computations compared to greedy routing.
> Thank you for raising this important point. Below we address the request for a limitations and societal impact discussion.
> * Limitations:
>
> A practical limitation of our current approach is its reliance on a custom CUDA kernel for sparse attention, which hinders plug-and-play deployment in standard inference frameworks. To address this, we will release a high-performance, open-source implementation based on Triton, enabling broader reproducibility and easier integration.
>
> * Societal Impact:
>
> Our work advances **energy-efficient LLM inference**, which can reduce carbon footprint and hardware costs—particularly beneficial for edge devices and low-resource settings. However, as with any approximation technique, there is a **fidelity–efficiency trade-off**. Users must validate model behavior in high-stakes domains (e.g., healthcare, legal) before deployment. We emphasize that our method preserves semantic integrity better than pruning-only approaches, but caution remains warranted.
>
> We believe these revisions will significantly strengthen the paper’s clarity, rigor, and impact.

---

> > ### Author Rebuttal · Reviewer_a7Sg · 2026-04-04
> >
> > Thank you for the clarification regarding inference speed relative to SkipGPT. The rebuttal adequately addresses my concerns about the LFF's presence in the critical path, and provides convincing evidence that it does not adversely impact inference speed.
> >
> > I also appreciate the conceptual contribution of the work. The shift from a binary execute-or-skip paradigm to a execute-or-approximate framework is both novel and well-motivated with respect to higher granularity of approximate computing, offering a meaningful perspective on token-level adaptive computation.
> >
> > Given these clarifications, I am inclined to raise my score.

---

> > > ### Author Response · Authors · 2026-04-04
> > >
> > > Thanks for the kind words! We're happy the explanation about inference speed made sense, and we really appreciate your support in raising the score.

---

### Official Review · Reviewer_kdxx · 2026-03-17

**Soundness:** 3
**Presentation:** 3
**Significance:** 3
**Originality:** 2
**Overall Recommendation:** 4
**Confidence:** 4

**Summary:**

This paper focused on dynamic computation allocation strategy and especially proposed informed routing that replaces the simple binary routing choice. It leverages lightweight feature forecaster (LFF), which approximates the input-output mapping of computation unit before routing decisions. This outperforms a few DCA prior works in terms of perplexity and few-shot accuracy.

**Compliance With Llm Reviewing Policy:**

Affirmed.

**Final Justification:**

Due to unavoidable circumstances as I stated in the official comment, I was unable to participate in the rebuttal discussion, and I sincerely apologize to the authors for this. Nevertheless, the concerns I raised reflect what I considered the most critical issues of this submission. I kindly ask the AC to carefully check whether these concerns have been adequately addressed in the authors' rebuttal. If not, I would maintain my current rating.

**Key Questions For Authors:**

See above weaknesses sections.

**Limitations:**

See above weaknesses sections.

**Strengths And Weaknesses:**

Strengths:

- The paper is well written and organized in overall. It was easy to follow the paper.
- The proposed method (LFF architecture and the concept) is straightforward, yet shows its effectiveness.
- The performance improvement over prior works like MoD or SkipGPT seems quite huge.

Weaknesses:

- Are there any results like in-context retrieval (long-context benchmarks) or generation benchmarks? I believe those results can correctly validate the proposed methods to check if they don't compromise the actual performance against dense models.
- LFF seems like they are compressing the given computational unit. So I'm not quite surprised they could outperform other baselines (the performance improvmenet over other methods seems good though). And I think the real issue is how much better they actually are compared to static methods (accuracy or speed). But overall, it seems like the paper focuses too much on comparing to SkipGPT.
- In Section 4.3, LFF looks like reducing Attention modules more. Do you think this reduction behavior would be good eventually? In terms of efficiency, it can be good. But in longer sequences or more smaller scales (where this kind of DCA techniques is required more), can losing attention parameters result in huge performance drop more?

---

> ### Author Rebuttal · Authors · 2026-03-28
>
> We sincerely thank the reviewer for their thoughtful comments and constructive feedback. In what follows, we address each of the concerns raised in the review point by point.
>
> ## Q1: Validation on long-context and generation benchmarks
> To rigorously evaluate robustness, we conducted new experiments on **LongBench (ACL 2024)** using **Llama3.2-3B** at 25% sparsity, covering in-context retrieval, long-code completion, and few-shot learning tasks:
>
> - **Long Code Completion (lcc_e)**
> - **Passage Retrieval (passage_retrieval_e)**
> - **Few-shot Learning (trec_e)**
>
> The results are summarized in the table below:
>
> | Task  / Metric                        | LLama3.2-3B | Greedy Routing | Informed Routing (ours) |
> |--------------------------------|----------|-----------------------------------------|------------|
> | Long Code Completion (lcc_e) / code_sim_score   | 0.2969  | 0.2427                                 | **0.2515**    |
> | Passage Retrieval (passage_retrieval_e) / retrieval_score | 0.1548  | 0.1128                                 | **0.1159**    |
> | Few-shot Learning (trec_e) / classification_score     | 0.6267  | 0.4800                                 | **0.5533**   |
>
> **Observations and Analysis:**
> * **our method consistently outperforms traditional Greedy Routing approaches** across long-context tasks, e.g. 55.33% accuracy vs. 48.00% on trec_e, aligning with the core conclusions from standard experiments.
>
> * Both sparse methods exhibit a slight performance drop compared to the dense baseline . We attribute this primarily to **distributional mismatch**: router and LoRA were trained on RedPajama (natural language, ≤2K tokens), while LongBench involves sequences up to 8K tokens and specialized domains (code, retrieval). This domain/length gap affects absolute performance but **does not diminish the relative superiority** of our approach.
>
> ## Q2: Comparison with static pruning methods
> We thank the reviewer for highlighting the importance of contextualizing our method within broader efficiency paradigms. We clarify the landscape as follows:
>
> * Dynamic Computation Allocation (DCA): Dynamically decides, based on each token’s representation, whether to skip the subsequent computational unit (e.g., an entire layer or a sub-layer such as an attention or FFN module).
> * Static Depth Pruning: Directly removes certain layers (or, in some recent works, sub-layers) from the network.
> * Static Width Pruning: Prunes neurons/channels, thereby reducing the intermediate feature dimensions.
>
> We emphasize a key conceptual insight: **DCA can be viewed as token-wise depth pruning**. In other words, static depth pruning is thus a constrained form of DCA, where all tokens must make identical skip/keep decisions per layer. Consequently:
>
> * **DCA has a strictly higher theoretical performance upper bound than static depth pruning**, as it subsumes the latter as a special case. It is also empirically confirmed by our experiments, e.g. for LLama3.1-8B under 40% sparsity,  the accuracy gap exceeded 19 points (53.37% vs. 72.43%) on arc-e.
>
> * That said, static pruning has advantages in terms of memory efficiency. When all tokens choose to skip a particular layer, the parameters of that layer can be entirely discarded. In contrast, DCA always requires storing the full set of model parameters, since any layer might still be used by some tokens.
>
> * Importantly, DCA is orthogonal to width pruning. Combining both could yield further efficiency gains—a direction we plan to explore in future work.
> ## Q3: Attention module skipping behavior and robustness
> We fully understand the reviewer’s concern: skipping more attention modules may weaken token–token interaction (since MLP does not model inter-token relations) and thus degrade performance. After careful analysis, we conclude that this is **not a fundamental limitation**, supported by the following evidence:
>
> 1. **Representation stability analysis**:
>    Using cosine similarity (scale-invariant under RMSNorm), we measured token representation shifts after LFF fitting:
>    - Self-attention: 0.91 (original) → 0.94 (LFF)
>    - FFN: 0.86 (original) → 0.89 (LFF)
>    → Attention induces *smaller* representation changes; skipping it preserves larger-magnitude transformations (FFN), aligning with routing efficiency principles.
>
> 2. **Token-type awareness**:
>    Router decisions correlate with linguistic role (Table 4): higher skip rates for function words (e.g., "and", "in") versus content words (nouns/verbs). Since function words contribute minimally to semantic interaction, reduced attention processing has negligible impact—consistent with token pruning literature [1].
>
> 3. **Adaptive routing balance**:
>    The router learns sparsity-dependent ratios *without explicit bias*:
>    - 25% sparsity: 71.4% attention skips vs. 28.6% FFN skips
>    - 70% sparsity: 66.2% vs. 33.8%
>    This demonstrates context-aware resource allocation.
>
>  Revisions will be implemented soon.

---

### Decision · Program_Chairs · 2026-04-30

**Decision:**

Accept (regular)

**Comment:**

This paper introduces an improved mechanism to learning a Lightweight Feature Forecaster to ensure dynamic computation through token selection is done before router. The reviewers agree the paper is well written and the idea is simple. I also agree with the reviewers assesment on figuring out large-scale batching to ensure dynamic compute is well served -- including continuous batching. I strongly urge the authors to investigate on that front further. The paper adds value to the community.